# Learning Fair Graph Representations via Automated Data Augmentations

**Hongyi Ling, Zhimeng Jiang, Youzhi Luo, Shuiwang Ji,** *Na Zou*\*

Texas A&M University
College Station, TX 77843, USA
{hongyiling,zhimengj,yzluo,sji,nzou1}@tamu.edu

## Abstract

We consider fair graph representation learning via data augmentations. While this direction has been explored previously, existing methods invariably rely on certain assumptions on the properties of fair graph data in order to design fixed strategies on data augmentations. Nevertheless, the exact properties of fair graph data may vary significantly in different scenarios. Hence, heuristically designed augmentations may not always generate fair graph data in different application scenarios. In this work, we propose a method, known as Graphair, to learn fair representations based on automated graph data augmentations. Such fairness-aware augmentations are themselves learned from data. Our Graphair is designed to automatically discover fairness-aware augmentations from input graphs in order to circumvent sensitive information while preserving other useful information. Experimental results demonstrate that our Graphair consistently outperforms many baselines on multiple node classification datasets in terms of fairness-accuracy trade-off performance. In addition, results indicate that Graphair can automatically learn to generate fair graph data without prior knowledge on fairness-relevant graph properties. Our code is publicly available as part of the DIG package (https://github.com/divelab/DIG).

## 1 Introduction

Recently, graph neural networks (GNNs) attract increasing attentions due to their remarkable performance (Gao et al., 2021; Gao & Ji, 2019; Liu et al., 2021a;b; Yuan et al., 2021) in many applications, such as knowledge graphs (Hamaguchi et al., 2017), molecular property prediction (Liu et al., 2022; 2020; Han et al., 2022a) and social media mining (Hamilton et al., 2017). Despite recent advances in graph representation learning (Grover & Leskovec, 2016; Kipf & Welling, 2017; 2016; Gilmer et al., 2017; Han et al., 2022b), these GNN models may inherit or even amplify bias from training data (Dai & Wang, 2021), thereby introducing prediction discrimination against certain groups defined by sensitive attributes, such as race and gender. Such discriminative behavior may lead to serious ethical and societal concerns, thus limiting the applications of GNNs to many real-world high-stake tasks, such as criminal justice (Suresh & Guttag, 2019), job hunting (Mehrabi et al., 2021), healthcare (Rajkomar et al., 2018), and credit scoring (Feldman et al., 2015; Petrasic et al., 2017). Hence, it is highly desirable to learn fair graph representations without discriminatory biases (Dong et al., 2022; Zhang et al., 2022; Kang et al., 2022; Dai et al., 2022).

A primary issue (Mehrabi et al., 2021; Olteanu et al., 2019) in fairness is that training data usually contain biases, which is the source of discriminative behavior of models. Thereby, many existing works (Agarwal et al., 2021; Kose & Shen, 2022; Spinelli et al., 2021) propose to learn fair graph representations by modifying training data with fairness-aware graph data augmentations. These methods propose some graph data properties that are beneficial to fair representation learning, and then adopt heuristic graph data augmentation operations, including node feature masking and edge perturbation, to refine graph data. However, the proposed graph properties (Spinelli et al., 2021; Kose & Shen, 2022) may not be appropriate for all graph datasets due to the diverse nature of graph data. For example, balanced inter/intra edges (Kose & Shen, 2022) may destroy topology

---

\*Equal senior contributions

structures of social networks, leading to the loss of important information. Even if the proposed graph properties are effective, the best graph properties may vary significantly in different scenarios. Hence, it is highly desirable to automatically discover dataset-specific fairness-aware augmentation strategies among different datasets with a single framework. To this end, a natural question is raised:

*Can we achieve fair graph representation learning via automated data augmentations?*

In this work, we attempt to address this question via proposing Graphair, a novel automated graph augmentation method for fair graph representation learning. A primary challenge is how to achieve fairness and informativeness simultaneously in the augmented data. As we intentionally avoid assuming prior knowledge on what types of graphs are considered fair, we propose to employ an adversary model to predict sensitive attributes from augmented graph data. A fair augmented graph should prevent the adversary model from identifying the sensitive attributes. In addition, we propose to retain useful information from original graphs by using contrastive learning to maximize the agreement between original and augmented graphs. Experimental results demonstrate that Graphair consistently outperforms many baselines on multiple node classification datasets in terms of fairness-accuracy trade-off performance.

## 2 BACKGROUND AND RELATED WORK

### 2.1 FAIR GRAPH REPRESENTATION LEARNING

In this work, we study the problem of fair graph representation learning. Let $\mathcal{G} = \{A, X, S\}$ be a graph with $n$ nodes. Here, $A \in \{0,1\}^{n \times n}$ is the adjacency matrix, and $A_{ij} = 1$ if and only if there exists an edge between nodes $i$ and $j$. $X = [x_1, \cdots, x_n]^T \in \mathbb{R}^{n \times d}$ is the node feature matrix, where each $x_i \in \mathbb{R}^d$ is the $d$-dimensional feature vector of node $i$. $S \in \{0,1\}^n$ is the vector containing sensitive attributes (e.g., gender or race) of nodes that should not be captured by machine learning models to make decisions. Our target is to learn a fair graph representation model $f : (A, X) \rightarrow H \in \mathbb{R}^{n \times d'}$, and the learned representation $H = f(A, X)$ is fed into a classification model $\theta : H \rightarrow \hat{Y} \in \{0,1\}^n$ to predict the binary label of nodes in $\mathcal{G}$. Particularly, for an ideal fair model $f$, the output representation $H$ should result in a prediction $\hat{Y}$ that satisfies the fairness criteria. In general, there exist several different definitions of fairness criteria, including group fairness (Dwork et al., 2012; Rahmattalabi et al., 2019; Jiang et al., 2022b), individual fairness (Kang et al., 2020; Dong et al., 2021; Petersen et al., 2021), and counterfactual fairness (Agarwal et al., 2021; Ma et al., 2022). In this work, we focus on group fairness, which is defined as

$$\mathbb{P}(\hat{Y}_i | S_i = 0) = \mathbb{P}(\hat{Y}_i | S_i = 1), \quad i = 1, \ldots, n, \tag{1}$$

where $\hat{Y}_i$ is the prediction for node $i$, and $S_i$ is the sensitive attribute of node $i$. Note that even though the sets of node attributes or features in $X$ and $S$ are disjoint, correlations may exist between $(A, X)$ and $S$. Hence, even if $S$ is not explicitly exposed to $f$, $f$ may implicitly infer parts of $S$ from $(A, X)$ and produce biased representation $H$, thereby making the prediction $\hat{Y}$ unfair. How to prevent models from intentionally fitting these correlations is the central problem to be solved in achieving fair graph representation learning.

Currently, several studies have proposed different strategies to achieve fair graph representation learning. An early study (Rahman et al., 2019) proposes to train the model through fair random walks. Some recent studies (Li et al., 2020; Laclau et al., 2021) propose to reduce prediction discrimination through optimizing adjacency matrices, which can improve fairness for link prediction tasks. In addition, adversarial learning is another popular strategy to achieve fairness on node representation learning tasks. Many studies (Fisher et al., 2020; Dai & Wang, 2021; Bose & Hamilton, 2019) adopt adversarial learning to filter out sensitive attribute information from the learned node representations. Overall, most existing methods learn fair representations via altering model training strategy with fairness regularization. However, a primary issue in fairness learning lies in the fact that training data usually possess bias. Hence, an alternative and highly desirable solution is to modify data through data augmentations, thus enabling models to learn fair representations easily. In this work, we design a learnable graph augmentation method to reduce bias in graph data, leading to more effective fairness-aware representation learning on graphs.

## 2.2 GRAPH DATA AUGMENTATIONS

Inspired by the success of data augmentations in computer vision and natural language processing, graph data augmentation (Zhao et al., 2022) attracts increasing attention in academia. Most studies (You et al., 2020; Zhu et al., 2020; Wang et al., 2021; Veličković et al., 2019; You et al., 2021; Rong et al., 2020) are based on uniformly random modifications of graph adjacency matrices or node features, such as masking node features, dropping edges, or cropping subgraphs. In addition, recent studies (Luo et al., 2023; Zheng et al., 2020; Luo et al., 2021; Zhao et al., 2021; Chen et al., 2020) design learnable data augmentation methods to enhance task-relevant information in augmented graphs. Note that none of the above methods are fairness-aware and only a few studies have investigated fairness-aware graph augmentations. Spinelli et al. (2021) argue that the tendency of nodes with the same sensitive attribute to connect leads to prediction discrimination. Thereby, they propose a biased edge drop algorithm to reduce such tendency in graphs, resulting in fairness improvement on prediction tasks. Agarwal et al. (2021) design a graph data augmentation method in the contrastive learning framework via modifying sensitive attributes. Kose & Shen (2022) study correlations between sensitive attributes and learned node representations, and propose several graph augmentations to minimize an upper bound of the correlations to achieve fairness. However, these fairness-aware augmentation methods are all based on some strong assumptions or definitions about the properties that fair graph data should have. Such assumptions or definitions may not hold in different scenarios, so in practice, empirical comparisons are needed to find out the best choice. In addition, these heuristic augmentation operations may accidentally remove most of the useful information from the graph. For instance, both edge drop algorithms proposed by Kose & Shen (2022) and Spinelli et al. (2021) may drop most of the edges and destroys the graph structure in some cases. Hence, in practice, these methods do not consistently achieve good performance on all datasets.

## 3 FAIRNESS VIA AUTOMATED DATA AUGMENTATIONS

While previous fairness-aware graph data augmentations all rely on manually defined and fixed fairness-relevant augmentation strategies, we explore a more adaptive and effective method to discover fairness-aware graph augmentations by automated augmentation models. Note that though automated graph augmentations have been applied to some graph representation tasks (Luo et al., 2023; 2021; Zhao et al., 2021), they have not been studied in fair graph representation learning. In this work, we propose Graphair, an automated graph augmentation method for fair graph representation learning. Graphair uses an automated augmentation model to generate new graphs with fair topology structures and node features while preserving the most informative components from input graphs. The augmentation model is trained end-to-end with multiple optimization objectives in order to circumvent sensitive information while retaining other useful information simultaneously. To the best of our knowledge, Graphair is the first automated graph augmentation method addressing group fairness with a theoretical guarantee of fairness and informativeness.

### 3.1 AUTOMATED GRAPH AUGMENTATIONS

We first present the details of the augmentation process. Given an input graph $\mathcal{G} = \{A, X, S\}$, we use the automated augmentation model $g$ to generate a new graph $\mathcal{G}' = \{A', X', S\}$ as

$$T_A, T_X = g(A, X), \quad A' = T_A(A), \quad X' = T_X(X). \tag{2}$$

Here, $T_A$ is the edge perturbation transformation, which maps $A$ to the new adjacency matrix $A'$ by removing existing edges and adding new edges. $T_X$ is the node feature masking transformation, which produces the new node feature matrix $X'$ by setting some values of $X$ to zero. $T_A$ and $T_X$ contain the exact transformations for each edge and node feature in $\mathcal{G}$. In other words, the augmentation model $g$ decides whether there is an edge connecting any two nodes in $\mathcal{G}$ and whether each value in $X$ should be set to zero or not.

In the augmentation model $g$, a GNN-based augmentation encoder $g_{\text{enc}} : (A, X) \to Z \in \mathbb{R}^{n \times d_r}$ is first used to extract $d_r$-dimensional embeddings $Z$ for nodes in $\mathcal{G}$. We adopt graph convolutional network (GCN) (Kipf & Welling, 2017) as the GNN encoder here. Afterward, the exact transformations for each edge and node feature are performed as described below.

**Edge perturbation.** Given the embedding $Z$, an multi-layer perceptron (MLP) model $\text{MLP}_A$ first computes the hidden embeddings $Z_A \in \mathbb{R}^{n \times d_{r'}}$ from $Z$, then an inner-product decoder computes the

edge probability matrix $\widetilde{A'} \in \mathbb{R}^{n \times n}$, where the value $\widetilde{A'}_{ij}$ at the $i$-th row, $j$-th column of the matrix $\widetilde{A'}$ denotes the predicted probability that an edge exists between the nodes $i$ and $j$ in $\mathcal{G'}$. Finally, the output adjacency matrix $A'$ is obtained by sampling from the Bernoulli distribution parameterized with the probabilities in $\widetilde{A'}$. Formally, this process can be described as

$$Z_A = \text{MLP}_A(Z), \quad \widetilde{A'} = \sigma\left(Z_A Z_A^T\right), \quad A'_{ij} \sim \text{Bernoulli}\left(\widetilde{A'}_{ij}\right) \text{ for } i, j = 1, \cdots, n, \quad (3)$$

where $\sigma(\cdot)$ is the sigmoid function.

**Node feature masking.** Given the embedding $Z$, an MLP model $\text{MLP}_X$ first computes the mask probability matrix $\widetilde{M} \in \mathbb{R}^{n \times d}$, where the value $\widetilde{M}_{ij}$ at the $i$-th row, $j$-th column of the matrix $\widetilde{M}$ denotes the predicted probability that the $j$-th feature of node $i$ is not set to zero. Afterward, the mask matrix $M$ is sampled from the Bernoulli distribution parameterized with the probabilities in $\widetilde{M}$, and the new feature matrix $X'$ is obtained by multiplying $X$ by $M$. This process can be formally described as

$$Z_X = \text{MLP}_X(Z), \, \widetilde{M} = \sigma(Z_X), \, M_{ij} \sim \text{Bernoulli}\left(\widetilde{M}_{ij}\right) \text{ for } i, j = 1, \cdots, n, \, X' = M \odot X, \, (4)$$

where $\odot$ is the Hadamard product, and $\sigma(\cdot)$ is the sigmoid function.

Note that the Bernoulli sampling for adjacency matrix $A'$ and mask matrix $M$ are non-differentiable. To make the augmentation model $g$ end-to-end trainable, we adopt the commonly-used trick to approximate the Bernoulli sampling in Eq. (3) and (4). Specifically, we relax the Bernoulli sampling procedure by the Gumbel-Softmax reparameterization trick (Jang et al., 2017; Maddison et al., 2017; 2014). Given a probability $\widetilde{P}$ computed from a parameterized model $\varphi$, the relaxed Bernoulli sampling calculates a continuous approximation $\hat{P} = \frac{1}{1+\exp(-(\log \widetilde{P}+G)/\tau)}$, where $\tau$ is a temperature hyperparameter and $G \sim \text{Gumbel}(0, 1)$ is a random variable sampled from the standard Gumbel distribution. For the forward propagation, the discrete value $P = \lfloor \hat{P} + \frac{1}{2} \rfloor$ is used as the result sampled from the Bernoulli distribution with the probability $\widetilde{P}$. For the backward propagation, a straight-through gradient estimator (Bengio et al., 2013) is used, which approximates the gradient as $\nabla_\varphi \hat{P} \approx \nabla_\varphi P$.

## 3.2 ADVERSARIAL TRAINING

As our objective is to generate fair augmentations to reduce bias, the ideal augmentation model $g$ should satisfy the **fairness** property. In other words, it should assign low probabilities to graph elements (edges, node features) that cause prediction bias. However, we cannot achieve it via supervised training because there is no ground truth indicating which graph elements lead to prediction bias and should be modified. To tackle this issue, we propose to use an adversarial learning based method to implicitly optimize the model to learn to mitigate bias in the input graph. Specifically, we use an adversary model $k : (A', X') \rightarrow \hat{S} \in [0, 1]^n$ to predict the sensitive attribute $S$ from the new adjacency matrix $A'$ and new node feature matrix $X'$ generated by the augmentation model $g$. The adversary model $k$ and the augmentation model $g$ are jointly trained via an adversarial fashion. In this process, $k$ is optimized to maximize the prediction accuracy of the sensitive attribute, while $g$ is optimized to mitigate bias in $A'$ and $X'$ so that it is difficult for the adversary model $k$ to identify sensitive attribute information from $A'$ and $X'$. Formally, this adversarial training process can be described as the following optimization problem:

$$\min_g \max_k L_{\text{adv}} = \min_g \max_k \frac{1}{n} \sum_{i=1}^n \left[ S_i \log \hat{S}_i + (1 - S_i) \log \left(1 - \hat{S}_i\right) \right], \quad (5)$$

where $\hat{S}_i$ is the prediction of the sensitive attribute of node $i$ by the adversary model $k$. [1]

## 3.3 CONTRASTIVE TRAINING

We note that only using the adversarial training may cause the augmentation model $g$ to collapse into trivial solutions. For instance, $g$ may learn to always generate a complete graph and set all node

---

[1]Here we use negative binary cross-entropy loss, so the adversary model $k$ aims to maximize $L_{\text{adv}}$.

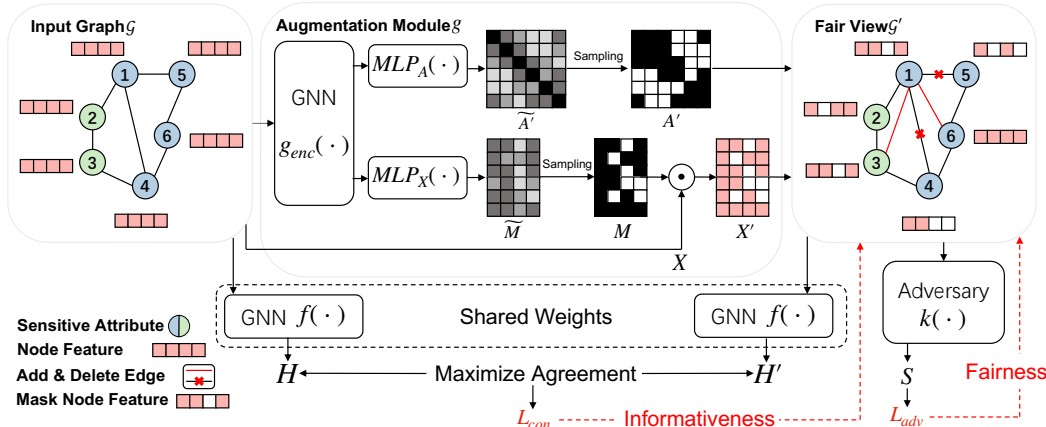

Figure 1: An overview of our framework.

features to zero, which contains no bias, since all nodes are equivalent. Such augmented graphs are not informative at all because they lose all the information from the input graphs. To make the augmentation model $g$ satisfy the **informativeness** property, i.e., preserving the most informative components of the input graph in the generated graphs, we additionally use a contrastive learning objective during training.

Given the input graph $\mathcal{G} = \{A, X, S\}$ and the augmented graph $\mathcal{G}' = \{A', X', S\}$, we first use a GNN-based representation encoder $f$ to extract node representations $H = f(A, X)$ and $H' = f(A', X')$ from $\mathcal{G}$ and $\mathcal{G}'$, respectively. Afterward, we optimize the augmentation model $g$ and the representation encoder $f$ jointly by minimizing a contrastive objective, which maximizes the similarity between the representations of the same node in $H$ and $H'$. Specifically, let $h_i$ and $h'_i$ denote the representation of node $i$ in $H$ and $H'$, respectively. For node $i$, we consider $(h_i, h'_i)$ as a positive pair, and $(h_i, h_j)$ and $(h_i, h'_j)$ for any node $j$ other than $i$ as negative pairs. We define the representation similarity as $\text{sim}(h_i, h'_j) = c(t(h_i), t(h'_j))$, where $c$ is the cosine similarity and $t$ is a non-linear projection implemented with a two-layer MLP model. We follow Zhu et al. (2020) to define the contrastive objective for any positive pair $(h_i, h'_i)$ as

$$l(h_i, h'_i) = -\log \frac{\exp\left(\text{sim}(h_i, h'_i)/\tau\right)}{\sum_{j=1}^{n} \exp\left(\text{sim}(h_i, h'_j)/\tau\right) + \sum_{j=1}^{n} \mathbb{1}_{[j \neq i]} \exp\left(\text{sim}(h_i, h_j)/\tau\right)}, \quad (6)$$

where $\tau$ denotes the temperature parameter, $\mathbb{1}_{[j \neq i]} \in \{0, 1\}$ is the indicator function whose value is 1 if and only if $j \neq i$. The overall contrastive objective is computed over the positive pairs $(h_i, h'_i)$ and $(h'_i, h_i)$ for all nodes as

$$L_{\text{con}} = \frac{1}{2n} \sum_{i=1}^{n} \left[ l(h_i, h'_i) + l(h'_i, h_i) \right]. \quad (7)$$

To prevent the augmentation model $g$ from generating graphs that deviate too much from input graphs, we add a reconstruction-based regularization term to the overall training objective. Specifically, let $L_{\text{BCE}}$ and $L_{\text{MSE}}$ denote binary cross-entropy loss and mean squared error loss, respectively, and the regularization term is defined as

$$
\begin{aligned}
L_{\text{reconst}} &= L_{\text{BCE}}(A, \widetilde{A'}) + \lambda L_{\text{MSE}}(X, X') \\
&= -\sum_{i=1}^{n} \sum_{j=1}^{n} \left[ A_{ij} \log\left(\widetilde{A'}_{ij}\right) + (1 - A_{ij}) \log\left(1 - \widetilde{A'}_{ij}\right) \right] + \|X - X'\|_F^2, \quad (8)
\end{aligned}
$$

where $\lambda$ is a hyperparameter, and $\|\cdot\|_F$ denotes the Frobenius norm of matrix (Golub & Van Loan, 1996).

To sum up, the overall training process can be described as the following min-max optimization procedure,

$$\min_{f,g} \max_{k} L = \min_{f,g} \max_{k} \alpha L_{\text{adv}} + \beta L_{\text{con}} + \gamma L_{\text{reconst}}, \quad (9)$$

where $\alpha$, $\beta$, $\gamma$ are hyperparameters. The parameters of augmentation model $g$, adversary model $k$, and representation encoder $f$ are jointly optimized with this min-max optimization procedure. In each training step, we first update the parameters of $f$ and $g$ to minimize $L$ while keeping $k$ fixed, then update the parameters of $k$ to maximize $L_{\text{adv}}$ while keeping $f$ and $g$ fixed. See Figure 1 for an overview of our proposed Graphair method. The training algorithm is summarized in Appendix B.

## 3.4 DISCUSSIONS

Graphair learns different fairness-aware augmentation strategies for different graph datasets by the automated augmentation model, thereby eliminating the negative effect of fixed fairness-relevant augmentation strategies (Spinelli et al., 2021; Agarwal et al., 2021; Kose & Shen, 2022). In addition, Graphair mitigates bias by modifying both graph topology structures and node features, while some existing studies (Spinelli et al., 2021) only consider one of them. We demonstrate these advantages through extensive empirical studies in Section 4.2 and 4.3. Furthermore, we show in Section 3.5 and 3.6 that the used training objectives can be theoretically proven to help the augmentation model generate new graphs with fair topology structures and node features, and preserve the most informative components from the input graph simultaneously. Specifically, we use adversarial and contrastive learning to optimize the augmentation model to satisfy the **fairness** and **informativeness** properties, respectively.

## 3.5 THEORETICAL ANALYSIS OF FAIRNESS

Following Madras et al. (2018), we quantify the unfairness of a classifier $d : (A', X') \rightarrow [0,1]^n$ using demographic parity distance. Given a graph $\mathcal{G}' = (A', X', S)$, let $\hat{Y} = d(A', X') \in [0,1]^n$ denote the prediction of the classifier $d$ and $\hat{Y}_i$ is the prediction of node $i$. The demographic parity distance is defined as $\Delta_{DP}(d) \triangleq |\mathbb{E}_{i \sim S^0}(\hat{Y}_i) - \mathbb{E}_{j \sim S^1}(\hat{Y}_j)|$, where $S^0$ and $S^1$ denote the set of nodes whose sensitive attributes are 0 and 1, respectively. Note that $\Delta_{DP}(d) = 0$ if $\hat{Y} \perp S$, i.e., the group fairness discussed in Section 2.1 is satisfied. The following theorem shows that minimizing the optimal adversarial loss is equivalent to minimizing the unfairness of the classifier $d$, so minimizing the performance of the adversary model can indeed encourage the augmentation model to generate fair graphs.

**Theorem 1.** *Let $\mathcal{G}'$, $k$, $S$ be defined as above. For any downstream task, we consider a classifier $d : (A', X') \rightarrow \hat{Y} \in [0,1]^n$ predicting label $Y \in \{0,1\}^n$ using $\mathcal{G}'$ as input. Assume the adversarial loss for each sample is bounded, i.e., there exists constant $M$ so that $|S_i log \hat{S}_i + (1 - S_i) log \left(1 - \hat{S}_i\right)| \leq M$ holds for each sample. Then we show that the demographic parity $\Delta_{DP}(d)$ is bounded by the optimal adversarial objective value $L_{adv}{}^*$, i.e., $L_{adv}{}^* \geq \frac{n'M}{n(1-e^M)}\Delta_{DP}(d) - \frac{n'M}{n(1-e^{-M})}$, where $n'$ represents the maximal number of samples with the same sensitive attributes.*

Detailed proof of this theorem is given in Appendix A.1.

## 3.6 THEORETICAL ANALYSIS OF INFORMATIVENESS

We quantify the amount of information obtained about one random variable by observing the other random variable by mutual information. We show in the following theorem that minimizing the contrastive loss $L_{\text{con}}$ is equivalent to maximizing a lower bound of the mutual information $I(\mathcal{G}; \mathcal{G}')$ between the original graph $\mathcal{G}$ and the augmented graph $\mathcal{G}'$, thus achieving **informativeness**.

**Theorem 2.** *Let $\mathcal{G}, \mathcal{G}', H$ and $H'$ be defined as above. Our contrastive objective is a lower bound of mutual information between the input graph $\mathcal{G}$ and the augmented graph $\mathcal{G}'$. Formally,*

$$-L_{con} \leq I(\mathcal{G}; \mathcal{G}'). \tag{10}$$

Detailed proof of this theorem is given in Appendix A.2.

## 3.7 COMPLEXITY ANALYSIS

Graphair shares the same time and space complexity as the GNN architecture of the representation encoder $f$ during inference because only $f$ is used to compute node representations. During training,

Table 1: Comparisons between our method and baselines on node classification tasks in terms of accuracy and fairness. The best results are shown in bold.

| Method | NBA | | | Pokec-z | | | Pokec-n | | |
|---|---|---|---|---|---|---|---|---|---|
| | ACC $\uparrow$ | $\Delta_{DP}\downarrow$ | $\Delta_{EO}\downarrow$ | ACC $\uparrow$ | $\Delta_{DP}\downarrow$ | $\Delta_{EO}\downarrow$ | ACC $\uparrow$ | $\Delta_{DP}\downarrow$ | $\Delta_{EO}\downarrow$ |
| FairWalk | $64.54 \pm 2.35$ | $3.67 \pm 1.28$ | $9.12 \pm 7.06$ | $67.07 \pm 0.24$ | $7.12 \pm 0.74$ | $8.24 \pm 0.75$ | $65.23 \pm 0.78$ | $4.45 \pm 1.25$ | $4.59 \pm 0.86$ |
| FairWalk+$\mathbf{X}$ | $69.74 \pm 1.71$ | $14.61 \pm 4.98$ | $12.01 \pm 5.38$ | $69.01 \pm 0.38$ | $7.59 \pm 0.96$ | $9.69 \pm 0.09$ | $67.65 \pm 0.60$ | $4.46 \pm 0.38$ | $6.11 \pm 0.54$ |
| GRACE | $70.14 \pm 1.40$ | $7.49 \pm 3.78$ | $7.67 \pm 3.78$ | $68.25 \pm 0.99$ | $6.41 \pm 0.71$ | $7.38 \pm 0.84$ | $67.81 \pm 0.41$ | $10.77 \pm 0.68$ | $10.69 \pm 0.69$ |
| GCA | $\mathbf{70.43 \pm 1.19}$ | $18.08 \pm 4.80$ | $20.04 \pm 4.34$ | $\mathbf{69.34 \pm 0.20}$ | $6.07 \pm 0.96$ | $7.39 \pm 0.82$ | $67.07 \pm 0.14$ | $7.90 \pm 1.10$ | $8.05 \pm 1.07$ |
| FairDrop | $69.01 \pm 1.11$ | $3.66 \pm 2.32$ | $7.61 \pm 2.21$ | $67.78 \pm 0.60$ | $5.77 \pm 1.83$ | $5.48 \pm 1.32$ | $67.32 \pm 0.61$ | $4.05 \pm 1.05$ | $3.77 \pm 1.00$ |
| NIFTY | $69.93 \pm 0.09$ | $3.31 \pm 1.52$ | $4.70 \pm 1.04$ | $67.15 \pm 0.43$ | $4.40 \pm 0.99$ | $3.75 \pm 1.04$ | $65.52 \pm 0.31$ | $6.51 \pm 0.51$ | $5.14 \pm 0.68$ |
| FairAug | $66.38 \pm 0.85$ | $4.99 \pm 1.02$ | $6.21 \pm 1.95$ | $69.17 \pm 0.18$ | $5.28 \pm 0.49$ | $6.77 \pm 0.45$ | $\mathbf{68.61 \pm 0.19}$ | $5.10 \pm 0.69$ | $5.22 \pm 0.84$ |
| Graphair | $69.36 \pm 0.45$ | $\mathbf{2.56 \pm 0.41}$ | $\mathbf{4.64 \pm 0.17}$ | $68.17 \pm 0.08$ | $\mathbf{2.10 \pm 0.17}$ | $\mathbf{2.76 \pm 0.19}$ | $67.43 \pm 0.25$ | $\mathbf{2.02 \pm 0.40}$ | $\mathbf{1.62 \pm 0.47}$ |

Graphair computes the adjacency matrix $A'$ and the pairwise similarity in the contrastive loss, thus having a space complexity of $O(n^2)$, where $n$ is the number of nodes. Fortunately, we can easily adopt the graph sampling-based batch training method proposed by Zeng et al. (2020) to perform mini-batch training and reduce the space complexity to $O(m^2)$, where $m$ is the batch size. More details on mini-batch training are given in Appendix C.

## 4 EXPERIMENTS

In this section, we evaluate Graphair on three real-world datasets, including NBA, Pokec-z and Pokec-n [2]. More details on datasets are given in Appendix F.1. Experimental results show that Graphair outperforms many baselines on node classification tasks in terms of both fairness and accuracy. To gain insights from learned fair graph data, we provide a comprehensive analysis on learned fair graph topology structures and fair node features. Our analysis results are consistent with studies (Spinelli et al., 2021; Kose & Shen, 2022; Jiang et al., 2022a; Dai & Wang, 2021). We also provide runtime experiments and hyperparameter studies in Appendix D.

### 4.1 EXPERIMENTAL SETTINGS

**Evaluation metrics.** We use accuracy to evaluate prediction performance of node classification tasks. To quantify group fairness, we follow studies (Louizos et al., 2016; Beutel et al., 2017) to adopt demographic parity $\Delta_{DP} = |\mathbb{P}(\hat{Y} = 1|S = 0) - \mathbb{P}(\hat{Y} = 1|S = 1)|$ and equal opportunity $\Delta_{EO} = |\mathbb{P}(\hat{Y} = 1|S = 0, Y = 1) - \mathbb{P}(\hat{Y} = 1|S = 1, Y = 1)|$, where $Y$ and $\hat{Y}$ denote ground-truth labels and predictions, respectively. Note that a model with lower DP and EO implies better fairness performance.

**Baselines.** We compare our methods with the following baseline methods, including (1) Fairwalk (Rahman et al., 2019), a fairness-aware random walk (Grover & Leskovec, 2016) for unsupervised node representation learning task; (2) GRACE (Zhu et al., 2020), deep graph contrastive representation learning with uniform random graph augmentations; (3) GCA (Zhu et al., 2021), graph contrastive learning with adaptive augmentations; (4) NIFTY (Agarwal et al., 2021), the first graph contrastive learning method with fairness-aware graph augmentations. Note that we only use the unsupervised component of NIFTY to learn node representations; (5) FairDrop (Spinelli et al., 2021), a heuristic edge dropping method to enhance fairness in graph representation learning; (6) FairAug (Kose & Shen, 2022), an adaptive data augmentation method for fair node representation learning. We adopt FairDrop and FairAug in the contrastive learning framework of GRACE to learn node representations, since they are both graph augmentation methods. The optimal hyperparameters for all methods are obtained by grid search.

**Evaluation protocol.** We use an evaluation protocol following Veličković et al. (2019) for a fair comparison. Specifically, we first learn the fair representation $H$ in a fully unsupervised manner as described in Section 3.2 and 3.3. Then the representation $H$ is used to train and test a simple

---

[2]We adopt the graph mini-batch training method proposed by Zeng et al. (2020) on Pokec-z and Pokec-n datasets.

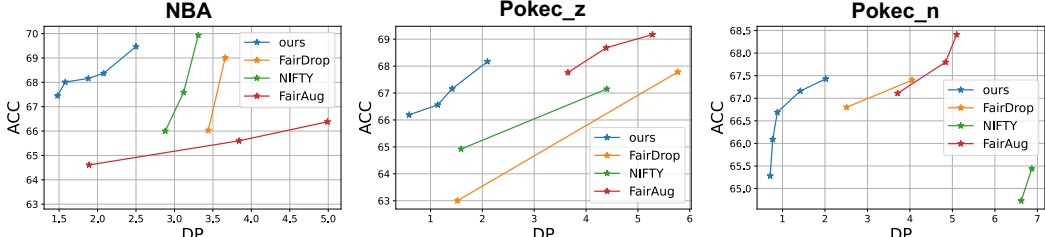

Figure 2: ACC and DP trade-off on three real-world datasets. Upper-left corner (high accuracy, low demographic parity) is preferable.

Table 2: Comparisons among different components in the augmentation model.

| Models | NBA | | | Pokec-z | | | Pokec-n | | |
|---|---|---|---|---|---|---|---|---|---|
| | ACC ↑ | $\Delta_{DP}$ ↓ | $\Delta_{EO}$ ↓ | ACC ↑ | $\Delta_{DP}$ ↓ | $\Delta_{EO}$ ↓ | ACC ↑ | $\Delta_{DP}$ ↓ | $\Delta_{EO}$ ↓ |
| Graphair w/o FM | 69.36 ± 0.28 | 4.95 ± 1.43 | 6.49 ± 1.78 | 67.04 ± 0.26 | 2.34 ± 0.34 | 3.99 ± 0.45 | 66.62 ± 0.18 | 2.36 ± 0.50 | 2.71 ± 0.41 |
| Graphair w/o EP | 66.67 ± 0.71 | 4.44 ± 2.64 | 6.74 ± 2.13 | 69.34 ± 0.40 | 5.75 ± 0.63 | 5.91 ± 0.66 | 68.59 ± 0.17 | 3.31 ± 0.57 | 3.71 ± 0.97 |
| Graphair | 69.36 ± 0.45 | **2.56 ± 0.41** | **4.64 ± 0.17** | 68.17 ± 0.08 | **2.10 ± 0.17** | **2.76 ± 0.19** | 67.43 ± 0.25 | **2.02 ± 0.40** | **1.62 ± 0.47** |

classifier. The test accuracy and fairness of this classifier are used as the proxy for the quality of the learned representation $H$.

## 4.2 EXPERIMENTAL RESULTS

**Fairness and accuracy performance.** Table 1 shows accuracy, demographic parity, and equal opportunity metrics of our proposed Graphair, compared with baselines in Section 4.1 on the three real-world datasets. From the results, we have the following observations:

- Our proposed Graphair consistently achieves the best fairness performance in terms of demographic parity and equal opportunity on evaluated datasets. For example, compared with GRACE, our method reduces demographic parity by 65.8%, 67.2% and 81.2% on NBA, Pokec-z, and Pokec-n datasets, respectively, with comparable accuracy performance.

- Fairness-aware augmentation methods (e.g., FairDrop, NIFTY, and FairAug) have lower prediction bias compared to GRACE and GCA. It is worth noting that these heuristic augmentation methods targeting manually designed fair graph properties may not consistently achieve state-of-the-art performance for all datasets due to diverse graph data. Specifically, FairDrop outperforms NIFTY on Pokec-n dataset, while NIFTY outperforms FairDrop on NBA and Pokec-z datasets. To this end, Graphair can automatically learn to discover fairness-aware augmentations on different graph datasets and thus outperforms all these fairness-aware methods in terms of demographic parity and equal opportunity on all three datasets.

**Trade-off between accuracy and fairness.** We further compare the accuracy-fairness trade-off performance of Graphair with several baselines. We choose demographic parity as the fairness metric. Figure 2 shows the Pareto front curves generated by a grid search of hyperparameters for each method. The upper-left corner point represents the ideal performance, i.e., highest accuracy and lowest prediction bias. The results show that Graphair achieves the best ACC-DP trade-off compared with all fairness-aware baselines on three datasets.

## 4.3 ABLATION STUDIES

Graphair considers two graph transformations to mitigate bias in node features and graph topology structures. In this subsection, we conduct ablation studies to investigate the contributions of two graph transformations and demonstrate the advances of Graphair. In other words, we investigate if both fair node features and graph topology structures enhance prediction fairness (i.e., lower DP and EO). Specifically, we remove node feature masking, denoted as "Graphair w/o FM", and remove edge perturbation, denoted as "Graphair w/o EP". Table 2 shows that Graphair outperforms both "Graphair w/o FM" and "Graphair w/o EP" in terms of demographic parity and equal opportunity on all three datasets. Experimental results demonstrate that both fair node features and graph topology

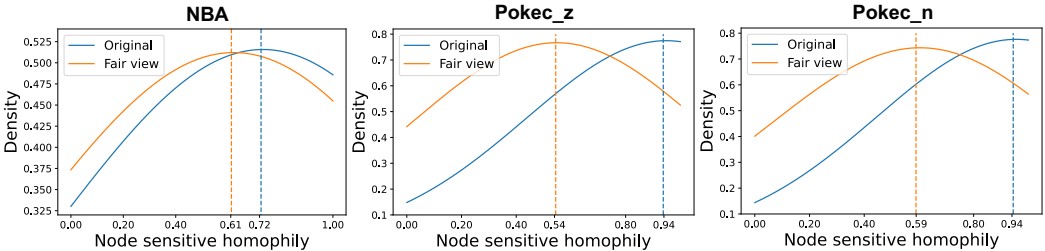

Figure 3: Node sensitive homophily distributions in the original and the fair graph data.

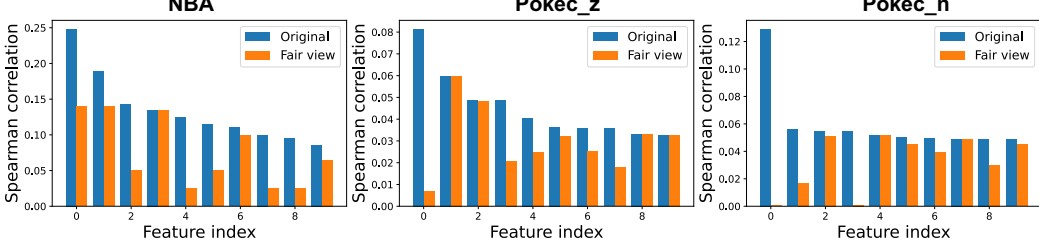

Figure 4: Spearman correlation between node features and the sensitive attribute in the original and the fair graph data.

structures are beneficial to mitigating prediction bias. Methods only considering either node features or graph topology (*e.g.*, FairDrop) are not promising due to the limited graph transformation space.

## 4.4 ANALYSIS OF FAIR VIEW

In this subsection, we study the properties of fair graph data generated by Graphair from graph topology and node features perspectives. Firstly, we introduce node-wise sensitive homophily coefficient to characterize the distribution of sensitive attributes from the neighborhood. Given a graph $\mathcal{G} = \{A, X, S\}$, node-wise sensitive homophily coefficient for node $i$, denoted as $\epsilon_i$, represents the proportion of neighbors with the same sensitive attributes, i.e., $\epsilon_i = \frac{\sum_{j=1}^{n} A_{ij} \mathbb{1}_{[s_i = s_j]}}{\sum_{j=1}^{n} A_{ij}}$, where $n$ is the number of nodes, and $\mathbb{1}_{[s_i = s_j]}$ is the indicator function evaluating to 1 if and only if $s_i = s_j$. Subsequently, we analyze the learned fair graph topology via node sensitive homophily distribution compared with the original graph topology. Figure 3 shows that the learned fair graph topology reduces average node sensitive homophily compared to the original graph topology. Such observation is consistent with several previous studies (Spinelli et al., 2021; Kose & Shen, 2022; Jiang et al., 2022a; Dai & Wang, 2021) that high node sensitive homophily values lead to prediction bias.

Additionally, we analyze the learned fair node features via Spearman correlation (Zwillinger & Kokoska, 1999) between the sensitive attribute and non-sensitive features. Note that fair node features should have low Spearman correlation values. Figure 4 shows the top-10 Spearman correlation values in the original graph data. We can see that the learned fair node features reduce Spearman correlation values compared to the original node features, thus preventing models from fitting the correlations as discussed in Section 2.1. These analysis results demonstrate that our method Graphair can automatically learn to generate fair graph data without prior knowledge of fairness-relevant graph properties.

## 5 CONCLUSIONS

In this work, we propose Graphair, an automated graph augmentation method for fair representation learning. Graphair uses an automated augmentation model to generate new graphs with fair topology structures and node features, while preserving the most informative components from input graphs. We adopt adversarial learning and contrastive learning to achieve fairness and informativeness simultaneously in the augmented data. Experimental results demonstrate that Graphair consistently outperforms state-of-the-art baselines on node classification tasks for real-world graph datasets in terms of fairness-accuracy trade-off performance. In the future, we would like to improve the efficiency of Graphair and extend Graphair to the case where only limited sensitive attribute information is available.

ACKNOWLEDGMENTS

The work was supported, in part, by NSF (IIS-1939716, IIS-1900990, and IIS-2006861), and Cisco Research. The views and conclusions in this paper are those of the authors and should not be interpreted as representing any funding agencies. We would also like to thank the helpful feedback from the anonymous reviewers.

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

## A    PROOFS OF THEOREMS

### A.1    PROOF OF THEOREM 1

*Proof.* Let $\hat{Y} = d(\mathcal{G}') \in [0, 1]^n$ denotes the prediction of the classifier $d$ and $\hat{Y}_i$ is the prediction of node $i$. Suppose without loss of generality that $\mathbb{E}_{i \sim S^1}[\hat{Y}_i] \leq \mathbb{E}_{j \sim S^0}[\hat{Y}_j]$. Then, we have

$$\Delta_{DP}(d) = \mathbb{E}_{i \sim S^0}[\hat{Y}_i] - \mathbb{E}_{j \sim S^1}[\hat{Y}_j] = \mathbb{E}_{i \sim S^0}[\hat{Y}_i - 1] + \mathbb{E}_{j \sim S^1}[-\hat{Y}_j] + 1 \qquad (11)$$

We assume the bounded adversarial loss of each sample, i.e., $|\log(\hat{S}_i)| \leq M$ for the sample with sensitive attribute $S = 1$, and $|\log(1-\hat{S}_i)| \leq M$ for the sample with sensitive attribute $S = 0$. Based on the concavity of $\log(\cdot)$ function and Jensen's inequality, for any $x \in [0, 1]$ with $|\log(x)| \leq M$, we have

$$\log(x) \geq \frac{M}{1 - e^{-M}}(x - 1) \qquad (12)$$

Note that higher prediction indicates lower sensitive attribute value since $\mathbb{E}_{i \sim S^1}[\hat{Y}_i] \leq \mathbb{E}_{j \sim S^0}[\hat{Y}_j]$, we consider an adversary model $\widetilde{k}$ that output the opposite of classifier $d$, i.e., $\widetilde{k}(\mathcal{G}') = 1 - d(\mathcal{G}')$. Let $\hat{S} = \widetilde{k}(\mathcal{G}')$ denote the prediction of the adversary model $\widetilde{k}$. Then, we have

$$
\begin{aligned}
L_{\text{adv}} &= \frac{1}{n}\sum_{i=1}^{n}[S_i\log\hat{S}_i + (1 - S_i)\log(1 - \hat{S}_i)] \\
&= \frac{|S^1|}{n}\mathbb{E}_{i \sim S^1}[\log(\hat{S}_i)] + \frac{|S^0|}{n}\mathbb{E}_{i \sim S^0}[\log(1 - \hat{S}_i)] \\
&= \frac{|S^1|}{n}\mathbb{E}_{i \sim S^1}[\log(1 - \hat{Y}_i)] + \frac{|S^0|}{n}\mathbb{E}_{i \sim S^0}[\log(\hat{Y}_i)] \\
&\overset{(a)}{\geq} \frac{n'}{n}\left[\mathbb{E}_{i \sim S^1}[\log(1 - \hat{Y}_i)] + \mathbb{E}_{i \sim S^0}[\log(\hat{Y}_i)]\right] \\
&\overset{(b)}{\geq} \frac{n'}{n}\left[\mathbb{E}_{i \sim S^1}\left(-\frac{M}{1 - e^{-M}}\hat{Y}_i\right) + \mathbb{E}_{i \sim S^0}\left[\left(\frac{M}{1 - e^{-M}}\hat{Y}_i - \frac{M}{1 - e^{-M}}\right)\right]\right] \\
&= \frac{n'M}{n(1 - e^{-M})}\left[-\mathbb{E}_{i \sim S^1}[\hat{Y}_i] + \mathbb{E}_{i \sim S^0}[\hat{Y}_i - 1]\right] \\
&= \frac{n'M}{n(1 - e^{-M})}\Delta_{DP}(d) - \frac{n'M}{n(1 - e^{-M})}, \qquad (13)
\end{aligned}
$$

where inequality (a) holds since $n' = \max(|S^0|, |S^1|)$, and inequality (b) holds due to equation (12). An optimal adversary model $k^*$ should do at least better than any arbitrary choice of $k$, thereby we have $L_{\text{adv}}^* \geq L_{\text{adv}} \geq \frac{n'M}{n(1-e^M)}\Delta_{DP}(d) - \frac{n'M}{n(1-e^{-M})}$. $\qquad \square$

### A.2    PROOF OF THEOREM 2

*Proof.* According to (Zhu et al., 2020), the contrastive objective $-L_{\text{con}}$ is a lower bound of the true mutual information between $H$ and $H'$, i.e.,

$$-L_{\text{con}} \leq I(H; H'). \qquad (14)$$

According to the data processing inequality, we have $I(U; V) \geq I(U; W)$ for a Markov chain $U \rightarrow V \rightarrow W$, where $U, V, W$ are random variables. The representations $H$ and $H'$ are extracted from the original graph $\mathcal{G}$ and the fair view $\mathcal{G}'$, thus $H, A, X, A', X', H'$ satisfying relationship $H \leftarrow (A, X) \rightarrow (A', X') \rightarrow (H')$. This relation is Markov equivalent to $H \rightarrow (A, X) \rightarrow (A', X') \rightarrow (H')$, since $H$ and $(A', X')$ are conditionally independent given $(A, X)$. The Markov chain leads to the following inequality,

$$I(H; H') \leq I(A, X; H') \leq I(A, X; A', X') \leq I(A, X, S; A', X', S) = I(\mathcal{G}; \mathcal{G}'). \qquad (15)$$

Combining Eq. 14 and 15, we have

$$-L_{\text{con}} \leq I(\mathcal{G}; \mathcal{G}'), \qquad (16)$$

which completes our proof. $\qquad \square$

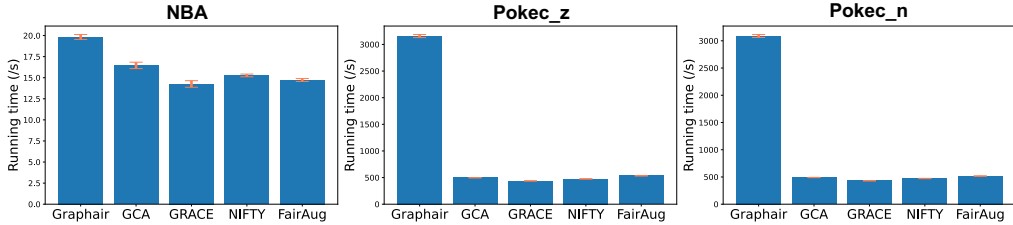

Figure 5: The running time comparison.

## B  TRAINING ALGORITHM FOR GRAPHAIR

We summarize the training algorithm for Graphair and provide the pseudo codes in Algorithm 1.

---

**Algorithm 1** Training algorithm

---

**Require:** adjacency matrix $A$, feature matrix $X$, sensitive attribute $S$
   **while** not converged **do**
       Generate a fair view $\mathcal{G}'$ using the augmentation model $g$
       Obtain node representations $H$ of $\mathcal{G}$ using the representation encoder $f$
       Obtain node representations $H'$ of $\mathcal{G}'$ using the representation encoder $f$
       Compute $L$ by Eq. (9)
       Update $f$ and $g$ by applying stochastic gradient descent to minimize $L$
       Update the adversary $k$ by applying stochastic gradient ascent to maximize $L$
   **end while**

---

## C  BATCH TRAINING FOR LARGE GRAPHS

Because Graphair has a space complexity of $O(n^2)$ in the full-batch setting, it is expensive to train Graphair on large graph datasets. To reduce space complexity, we adopt the graph sampling-based batch training method proposed by Zeng et al. (2020) to perform mini-batch training. Specifically, we construct a subgraph via a random walk sampler for each batch. Then the subgraph is used as the input of Graphair, and the augmentation model $g$ generates a fair view of the subgraph. Both adversarial training and contrastive training are performed on the subgraph. The normalization techniques in (Zeng et al., 2020) are also used to eliminate biases caused by subgraph sampling.

Note that the generated fair view for each batch might be subtly different from the one in the full-batch setting, because the augmentation model can only modify edges inside subgraphs in the mini-batch setting. Nevertheless, such a small difference won't make a big change when the batch size is large enough.

## D  MORE EXPERIMENTAL RESULTS

### D.1  RUNNING TIME COMPARISON

We provide the running time comparison in Figure 5 for our Graphair and baselines. We don't include FairWalk because the implementation we used doesn't use GPU to accelerate the training process. To achieve a fair comparison, we train all models for 500 epochs and report the average running time over 5 runs. When performing batch training on Pokec-z and Pokec-n datasets, we use a random walk sampler with 1000 root nodes and walk length 3. Note that FairAug proposes a fairness-aware graph sampling operation, so we use it to sample subgraphs with 3000 nodes instead of using a random walk sampler. Figure 5 shows that Graphair has a higher time complexity than other baselines. This is not surprising because all the baselines rely on fixed augmentation strategies and don't need a learnable neural network model.

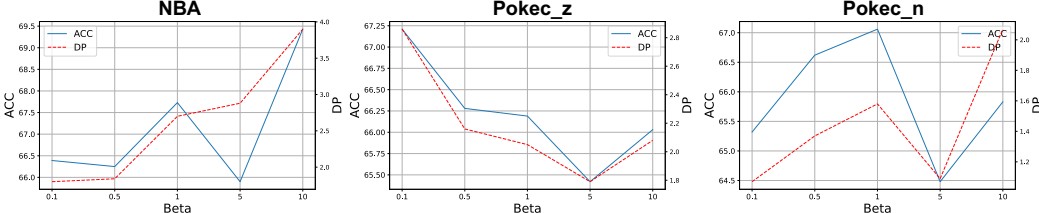

Figure 6: Ablation study on hyperparameter $\beta$.

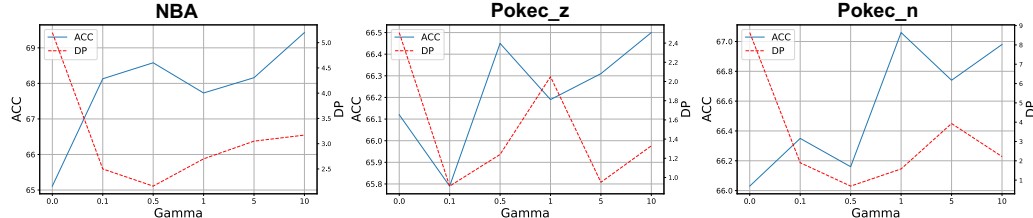

Figure 7: Ablation study on hyperparameter $\gamma$.

## D.2 HYPERPARAMETER STUDIES

In this subsection, we conduct hyperparameter studies for further investigation on the contribution of different components in Graphair. First, we tune hyperparameter $\beta$ among $\{0.1, 0.5, 1, 5, 10\}$. Note that the contrastive loss is unavoidable in training the representation encoder $f$, so we don't consider the case where $\beta = 0$. Results in Figure 6 show that change of $\beta$ leads to a trade-off between fairness and prediction performance. Additionally, we tune hyperparameter $\gamma$ among $\{0, 0.1, 0.5, 1, 5, 10\}$. Results in Figure 7 show that Graphair has a better performance (higher accuracy and lower demographic parity) with the reconstruction based regularization term ($\gamma \neq 0$) than without it ($\gamma = 0$). This is because the reconstruction loss can prevent the augmentation model $g$ from generating graphs that deviate too much from the input graph.

## E VISUALIZATION OF DIFFERENT AUGMENTATION METHODS

In this section, we provide a case study comparing different fairness-aware augmentation methods. In Figure 8, we show the change of a 1-hop ego graph in the NBA dataset. Node 1 is the ego node of this ego graph and other nodes are the 1-hop neighbors of Node 1. Results in Figure 8, show that FairDrop drops most edges in the original ego graph due to the high sensitive homophily of the ego node. In contrast, Graphair only drops one edge and preserves the original graph topology. Besides, Graphair reduces the sensitive homophily by connecting the ego node to a node with a different sensitive attribute (node 9).

## F MORE DETAILS ON EXPERIMENTAL SETTINGS

### F.1 DATASETS

We use three real-world social network datasets, including NBA, Pokec-z, and Pokec-n (Dai & Wang, 2021), to evaluate Graphair on node classification tasks. Pokec-z and Pokec-n are sampled from a larger social network Pokec, which is the most popular social network in Slovakia. User features contain gender, age, hobbies, interests, education, working field, etc. Among these features, region is treated as the sensitive attribute and working field is used as the predicted label. Since the node representations are learned in an unsupervised manner, we use a small portion of labeled data to train the classifier. In other words, we randomly split $10\%/10\%/80\%$ for training, validating and testing the classifier. NBA is extended from a Kaggle dataset with more than 400 NBA basketball players. The player information contains nationality, age, salary, performance statistics in the 2016-2017 season, etc. Nationality is treated as the sensitive attribute and the task is to predict if the salary of the player is over median. We randomly split $20\%/35\%/45\%$ for training, validating and

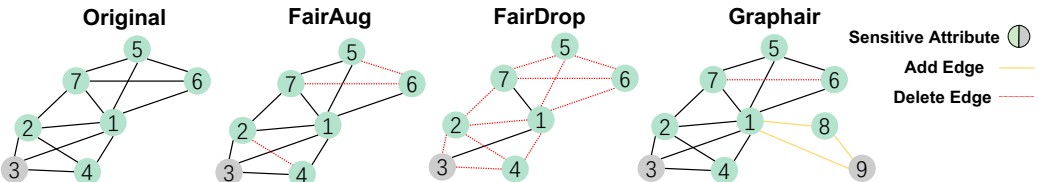

Figure 8: Comparison of different fairness-aware augmentation methods on the NBA dataset.

Table 3: The statistics of datasets.

| Dataset | NBA | Pokec-z | Pokec-n |
|---|---|---|---|
| # Nodes | 403 | 67,797 | 66,569 |
| # Node features | 39 | 59 | 59 |
| # Edges | 16,570 | 882,765 | 729,129 |
| # Inter-group edges | 4,401 | 39,804 | 31,515 |
| # Intra-group edges | 12,169 | 842,961 | 697,614 |

testing the classifier. The statistics of the datasets are given in Table 3. Note that we use the mini-batch training discussed in Appendix C on Pokec-z and Pokec-n datasets to reduce computation complexity.

### F.2 IMPLEMENTATION DETAILS

For Graphair, we adopt two-layer GCN models as the adversary model $k$ and augmentation encoder $g_{enc}$, and a three-layer GCN model as the representation encoder $f$. We use 64 as the hidden dimension in all three models. For the augmentation model, we use an MLP model with 2 layers, the hidden size of 64, and ReLU as the non-linear activation function for $MLP_A$ and $MLP_X$. The hyperparameter $\beta$ is set to 1, and the hyperparameters $\alpha, \gamma$ and $\lambda$ are determined with a grid search among $\{0.1, 1, 10\}$. For a fair comparison, we use three-layer GCN models for all baselines except FairWalk. The dimension of the node representations is selected as 64 for all datasets. We run the experiments 5 times and report the average performance for each method. We train the models for 500 epochs using Adam optimizer with $1 \times 10^{-4}$ learning rate and $1 \times 10^{-5}$ weight decay. For the results in Table 1, we select the optimal hyperparameters with the highest accuracy. For the classifier used for evaluation, we use an MLP model with 2 layers, the hidden size of 128, and ReLU as the non-linear activation function. The classifier is trained for 500 epochs using Adam optimizer with $1 \times 10^{-3}$ learning rate and $1 \times 10^{-5}$ weight decay.

## G EXPERIMENTS ON A SYNTHETIC GRAPH DATASET

To further validate the scalability of Graphair on larger graphs, we conduct experiments on a large synthetic graph dataset with 1,000,000 nodes. The synthetic dataset is generated as follows. We assume that the sensitive attribute is a binary value and randomly assign 0 or 1 to each node with equal probability. For node features, we use Gaussian Mixture Model to generate biased two-dimensional node features. The distributions of node features of different sensitive groups are different. Specifically, we use Gaussian distributions $\mathcal{N}(\mu_1, \Sigma)$ and $\mathcal{N}(\mu_2, \Sigma)$ to generate node features for nodes with sensitive attributes 0 and 1, respectively, where $\mu_1 = [0, 1]$, $\mu_2 = [1, 0]$ and $\Sigma = \begin{bmatrix} 1 & 0 \\ 0 & 2 \end{bmatrix}$.

For the adjacency matrix, we randomly generate edges via a stochastic block model. Since nodes with the same sensitive attributes are more likely to be connected in social networks, we generate edges with lower inter-connection and higher intra-connection probability between sensitive groups. Specifically, we set the probability of connecting two nodes with the same and different sensitive attributes as $1 \times 10^{-3}$ and $1 \times 10^{-4}$, respectively. For label generation, we intentionally make the labels correlated to the sensitive attributes. Specifically, for the label of each node, we set 0 as the threshold value and create a binary label based on the second dimension of the node features. Then we add noise to the labels by randomly flipping 20% of the labels to a different class. Since the node

Table 4: Comparisons between our method and baselines on the synthetic dataset in terms of accuracy and fairness. The best results are shown in bold.

| Models | Synthetic | | |
|---|---|---|---|
| | ACC $\uparrow$ | $\Delta_{DP} \downarrow$ | $\Delta_{EO} \downarrow$ |
| GRACE | $60.41 \pm 0.04$ | $47.06 \pm 4.20$ | $47.01 \pm 4.25$ |
| FairDrop | $59.05 \pm 9.47$ | $29.62 \pm 12.88$ | $29.66 \pm 11.88$ |
| NIFTY | $56.38 \pm 8.92$ | $29.24 \pm 12.16$ | $29.16 \pm 12.16$ |
| FairAug | $\mathbf{60.42 \pm 0.02}$ | $43.40 \pm 1.01$ | $43.33 \pm 1.00$ |
| Graphair | $60.36 \pm 0.02$ | $\mathbf{15.56 \pm 9.01}$ | $\mathbf{15.57 \pm 8.99}$ |

representations are learned in an unsupervised manner, we use a small portion of labeled data to train the classifier. In other words, we randomly split 10%/10%/80% for training, validating, and testing the classifier. We use the same hyperparameters for modeling training and the same architecture as discussed in Appendix F.2. We run the experiments 3 times and report the average performance for each method. We use the mini-batch training discussed in Appendix C on this large synthetic graph dataset.

Table 4 shows accuracy, demographic parity, and equal opportunity metrics of our proposed Graphair, compared with GRACE, FairDrop, NIFTY, and FairAug. From the results, fairness-aware augmentation methods have lower prediction bias compared to the uniform random augmentation method (i.e., GRACE). In addition, Graphair achieves the best fairness performance in terms of demographic parity and equal opportunity on this large synthetic dataset. These results demonstrate the scalability of our Graphair on large graph datasets.

