# OpenReview forum: "Learning Fair Graph Representations via Automated Data Augmentations"
_ICLR.cc/2023/Conference — ICLR 2023 notable top 25%_

### Official Review · Reviewer_QdJb · 2022-10-21

**Confidence:** 4
**Clarity, Quality, Novelty And Reproducibility:** 1. The key idea of the proposed metho…
**Correctness:** 3
**Technical Novelty And Significance:** 3
**Empirical Novelty And Significance:** 3
**Recommendation:** 8

**Strength And Weaknesses:**

[Strength]

1. This paper studies an important and practical problem.

2. The proposed solution is sound and solid.

3. The proposed method achieves SOTA performance.

4. The overall presentation is good.


[Weakness]

I have no significant concerns about the paper. However, I believe the experiment section can be improved.

1. Some details of experiments are missing. E,g., which model is used for the classifier?


2. The baselines can be strengthened by including SOTA method for fairness-aware graph learning, i.e., FairGNN [Dai & Wang, 2021].
Although FairGNN directly uses the class label in its training, I believe it can still be compared with the proposed method.
Also, I think that investigating the benefits of augmentation in cases where only a small portion of sensitive attributes is available (as done in [Dai & Wang, 2021]) can better support the significance of the proposed method.


**Summary Of The Paper:**

This paper proposes Graphair, a new framework for learning fair graph representations. Graphair consists of three major components: (1) automated graph augmentation, (2) adversarial training, (3) contrastive training. Experiments are conducted to show that the proposed framework achieves a better fairness-accuracy trade-off compared to the baseline methods.

**Summary Of The Review:**

In sum, this paper studies an important problem of learning fair graph representation, and the proposed framework provides a technically sound and solid solution.
Although the novelty of each proposed component is rather limited, the whole framework seems like an incrementing step forward in this research line. Further, experiment results support the superiority of the proposed framework.

---

> ### Author Response · Authors · 2022-11-12
> **Response to reviewer QdJb**
>
> We are very glad you had a positive initial impression and appreciate your constructive comments. We have revised the manuscript accordingly and provided pointwise responses below.
>
> >**Q1: Some details of experiments are missing. E,g., which model is used for the classifier?**
>
> We apologize for the missing details. For the classifier used for evaluation, we use an MLP model with 2 layers, the hidden size of 128, and ReLU as the
> non-linear activation function. We have added more implementation details in Appendix F.2 of the revised manuscript.
>
> >**Q2: The baselines can be strengthened by including SOTA method for fairness-aware graph learning, i.e., FairGNN. Investigating the benefits of augmentation in cases where only a small portion of sensitive attributes is available can better support the significance of the proposed method.**
>
> Thanks for this comment. **We believe that the problem in FairGNN is different from ours**. We consider fair graph representation learning via data augmentations, while FairGNN considers learning fair GNNs with limited sensitive attribute information. **There are two major differences.** First, we focus on learning fair graph representations, which is unsupervised learning. Instead, FairGNN studies the common node classification task, which is supervised learning. The other difference is that FairGNN considers the case where limited sensitive attribute information is available. FairGNN uses a GNN-based sensitive attribute estimator, which is specifically designed to address the lack of sensitive attribute information. In our work, we consider a different case that all sensitive attribute information is available. Although it is possible to extend our methods to solve limited sensitive information problems and compare with FairGNN, we believe it is out of the scope of this paper. In future work, we will study the case where only limited sensitive attribute information is available. We added the discussion of future work in Section 5 of the revised manuscript.

---

> > ### Comment · Reviewer_QdJb · 2022-11-25
> > **thank you**
> >
> > Thanks for the detailed response! I'll keep my recommendation.

---

### Official Review · Reviewer_oCtR · 2022-10-24

**Confidence:** 3
**Correctness:** 4
**Technical Novelty And Significance:** 3
**Empirical Novelty And Significance:** 3
**Recommendation:** 8

**Clarity, Quality, Novelty And Reproducibility:**

Overall I think this paper is well written and easy to read. I believe the ideas in the paper, though seem natural after being clarified in the paper, actually contain enough novelty.

**Strength And Weaknesses:**

I like this paper. Overall I see a well written paper with thorough theories/experiments provided. The fact that the augmentation method is not via some pre-determined rule is nice. The adoption of adversarial training for fairness property is also interesting (but for this part I'm not sure if similar things have appeared in other literatures or not).

The proofs of the two theorems are somewhat standard to me. This is not bad because I didn't see the claim on theoretical innovations. For experiments, the datasets used are relatively small. Maybe additional results on larger synthetic data will also be interesting together with discussions on the scalability of the proposed method.

**Summary Of The Paper:**

This paper studies fair graph representation learning via graph augmentation. Different from other literatures, this work proposed an automated augmentation methodology. Some theories on fairness and informativeness are provided, followed by experiments and ablation studies.

**Summary Of The Review:**

A well written paper with enough novelty. The fact that the augmentation method is not via some pre-determined rule is the interesting part.

---

> ### Author Response · Authors · 2022-11-12
> **Response to reviewer oCtR**
>
> We are very glad you like our paper and appreciate your constructive comments. We have revised the manuscript accordingly and also provide responses here.
>
> >**Q1: Maybe additional results on larger synthetic data will also be interesting together with discussions on the scalability of the proposed method.**
>
> Thanks for this constructive comment. As we mentioned in the Section 3.7, we can adopt the graph sampling-based batch training method proposed by [1] to perform mini-batch training on large graphs. The details of mini-batch training is provided in Appendix C.
>
> Additionally, we have conducted extra experiments on a synthetic graph dataset with 1000000 nodes. We use the Gaussian mixture model to generate node features. The adjacency matrix is generated via the stochastic block model with lower inter-connection and higher intra-connection probability between sensitive groups. The experiment results in the following table (Table 4 in the revised manuscript) show that Graphair can reduce prediction bias on large-scale graphs.
> The details of the experiments on the synthetic dataset are added in Appendix G of the revised manuscript.
>
> |Method|ACC|DP|EO|
> |----|----|----|----|
> |GRACE|60.41 $\pm$ 0.04|47.06 $\pm$ 4.20|47.01 $\pm$ 4.25|
> |FairDrop|59.05 $\pm$ 9.47|29.62 $\pm$ 12.88|29.66 $\pm$ 11.88|
> |NIFTY|56.38 $\pm$ 8.92|29.24 $\pm$ 12.16|29.16 $\pm$ 12.16|
> |FairAug|**60.42 $\pm$ 0.02**|43.40 $\pm$ 1.01|43.33 $\pm$1.00|
> |Graphair| 60.36 $\pm$ 0.02|**15.56 $\pm$ 9.01** |**15.57 $\pm$ 8.99**|
>
> > Reference
>
> [1] Zeng, H., Zhou, H., Srivastava, A., Kannan, R., & Prasanna, V. (2020). Graphsaint: Graph sampling based inductive learning method. International Conference on Learning Representations. In International Conference on Learning Representations.

---

> > ### Comment · Reviewer_oCtR · 2022-12-05
> > **Thanks for the response. I'll keep my recommendation.**
> >
> > Thanks for the response. I'll keep my recommendation.

---

### Official Review · Reviewer_fgyY · 2022-10-24

**Confidence:** 4
**Correctness:** 4
**Technical Novelty And Significance:** 3
**Empirical Novelty And Significance:** Not applicable
**Recommendation:** 8

**Clarity, Quality, Novelty And Reproducibility:**

The presentation of this paper is generally clear and easy to follow. The statements and theorems have been justified. As for originality, the paper includes necessary related works and how the proposed method is different from these recent works.
There are only minor issues in the presentation of results:
1. In Figure 8: it is a bit difficult to distinguish the color of original edges (black) and additional edges (green).
2. In Equations (3) and (4): Bernoulli distribution takes a single probability as the parameter, it would be better to express how to get A' and Z' in an elementwise form (e.g., $A'_{ij} =$).


**Strength And Weaknesses:**

Strengths:
1. The method proposed in the paper does not require some strong assumptions or definitions about the properties that fair graph data should have, while most existing methods rely on these assumptions which may vary significantly in different scenarios.

2. According to the experiments, the method achieves accuracy comparable to the state-of-the-art methods on node classification tasks on three datasets while significantly improves the fairness, and outperforms several baselines on the fairness-accuracy trade-off performance.

Weaknesses:
1. There are only limited results on how the method affects the graph topology. Apart from the case study presented in the appendix, more experimental results are expected on the comparison of changes in graph structure after the augmentations.

2. (This limitation has been addressed in the appendix. It would be better to also mention this in the main part.) Because Graphair has a space complexity of O(n^2) in the full-batch setting, it is expensive to train Graphair on large graph datasets, mini-batch is required to reduce space complexity.


**Summary Of The Paper:**

The paper proposes Graphair, an automated graph data augmentation method for fair graph representation learning. Graphair is designed to automatically discover fairness-aware augmentations from input graphs in order to circumvent sensitive information while preserving other informative features. Adversarial learning and contrastive learning are adopted to achieve fairness and informativeness simultaneously in the augmented graphs. Empirical results indicate that Graphair outperforms several recent baselines on node classification tasks in terms of fairness-accuracy trade-off performance.

**Summary Of The Review:**

The paper proposes a novel graph augmentation method for fair representation learning which outperforms the state-of-the-art methods on the fairness-accuracy trade-off performance. Although there are limitations on the complexity, the method is generally effective and novel in the field of fair graph representation. The presentation of materials is generally clear and the statements and theorems have been justified.

---

> ### Author Response · Authors · 2022-11-12
> **Response to reviewer fgyY**
>
> Thanks a lot for your constructive comments! We have revised the manuscript accordingly and also provide responses here.
>
> >**Q1: There are only limited results on how the method affects the graph topology. Apart from the case study presented in the appendix, more experimental results are expected on the comparison of changes in graph structure after the augmentations.**
>
> As we mentioned in Section 3.2, a primary challenge in fair graph representation learning is **the lack of ground truth indicating which graph elements (node features or edges) lead to prediction bias.** Thus, it is hard to analyze whether augmentation methods add/delete correct edges. **In addition to the case study presented in the appendix, we have analyzed changes in graph topology structure after the augmentations in Section 4.4.** We compared the changes in graph structure using node-wise sensitive homophily coefficient, which is the most commonly used fairness-related graph property. Many studies [1][2][3][4] point out that high node sensitive homophily values lead to prediction bias. As shown in Figure 3, Graphair reduces average node sensitive homophily, which is consistent with the previous studies.
>
> >**Q2: (This limitation has been addressed in the appendix. It would be better to also mention this in the main part.) Mini-batch is required to reduce space complexity.**
>
> We agree that it is expensive to train Graphair on large graph datasets. In the last two sentences of section 3.7, we mentioned that we adopt the graph sampling based batch training method to perform mini-batch training, thereby reducing the space complexity. We tried to put the batch training section in the main paper. However, due to the page limit, we have to put the details of  mini-batch training in Appendix C.
>
> **Minor issues:**
>
> >**In Figure 8: it is a bit difficult to distinguish the color of original edges (black) and additional edges (green).**
>
> We have updated the color of added edges to yellow. It should be better now.
>
> >**In Equations (3) and (4): Bernoulli distribution takes a single probability as the parameter, it would be better to express how to get A' and Z' in an elementwise form.**
>
> We have updated Equations (3) and (4) in the revised manuscript.
>
> > Reference
>
> [1] Dai, E., & Wang, S. (2021, March). Say no to the discrimination: Learning fair graph neural networks with limited sensitive attribute information. In Proceedings of the 14th ACM International Conference on Web Search and Data Mining (pp. 680-688).
>
> [2] Kose, O. D., & Shen, Y. (2022). Fair node representation learning via adaptive data augmentation. arXiv preprint arXiv:2201.08549.
>
> [3] Jiang, Z., Han, X., Fan, C., Liu, Z., Zou, N., Mostafavi, A., & Hu, X. (2022). Fmp: Toward fair graph message passing against topology bias. arXiv preprint arXiv:2202.04187.
>
> [4] Spinelli, I., Scardapane, S., Hussain, A., & Uncini, A. (2021). Fairdrop: Biased edge dropout for enhancing fairness in graph representation learning. IEEE Transactions on Artificial Intelligence, 3(3), 344-354.

---

> > ### Comment · Reviewer_fgyY · 2022-12-06
> > **Thanks for the response**
> >
> > The authors' response has addressed my concerns. I have raised the rating.

---

### Official Review · Reviewer_fjjw · 2022-10-24

**Confidence:** 4
**Correctness:** 3
**Technical Novelty And Significance:** 2
**Empirical Novelty And Significance:** 2
**Recommendation:** 6

**Clarity, Quality, Novelty And Reproducibility:**

Clarity and quality: this paper is overall clearly written and easy to follow.

Novelty: this work is fairly novel to me.

Reproducibility: the code/implementation was not provided.

**Strength And Weaknesses:**

s1. This paper is overall clearly written and easy to follow.

s2. Learning fair representations is a very important topic in graph machine learning. And from this paper, data augmentation seems to be a good methodology for it.

s3. From the experimental results (table 1 and figure 2), the proposed method is able to significantly improve on fairness without sacrificing too much on accuracy.

w1. The empirical results are not convincing enough. If no public datasets are available, maybe the authors can consider evaluate on some larger scale synthetic datasets. This would also help showing the scalability of the proposed method, especially considering the current datasets are pretty small with 60k nodes max.

w2. From Figure 5, the proposed method seem to be very low even on not-so-large datasets.

w3. I suggest the authors to use vector graphics for all figures in the paper.

**Summary Of The Paper:**

This paper proposed an automated data augmentation method for fair graph representation learning. This seems to be the first automated graph data augmentation method that focuses on learning fair representations on graph data. The authors designed the Graphair framework with four components: edge augmentation, feature augmentation, adversarial training, and contrastive training. From the experimental results, the proposed method seems able to significantly improve the fairness while not sacrificing too much on accuracy.

**Summary Of The Review:**

Based on my opinions above, I recommend boarderline accept.

---

> ### Author Response · Authors · 2022-11-12
> **Response to reviewer fjjw**
>
> Thanks a lot for your valuable comments! We have revised the manuscript accordingly and provided pointwise responses below.
>
> >**Q1: The empirical results are not convincing enough due to the small size of current datasets. If no public datasets are available, maybe the authors can consider evaluating on some larger-scale synthetic datasets.**
>
> To our best knowledge, our used datasets Pokec-z and Pokec-n are the largest graph datasets with sensitive attributes. They have been used in previous works [1][2][3][4]. Unfortunately, there is no larger real-world graph dataset with sensitive attributes yet.
>
> Additionally, we have conducted extra experiments on a synthetic graph dataset with 1000000 nodes. We use the Gaussian mixture model to generate node features. The adjacency matrix is generated via the stochastic block model with lower inter-connection and higher intra-connection probability between sensitive groups. The experiment results in the following table (Table 4 in the revised manuscript) show that Graphair can reduce prediction bias on large-scale graphs.
> The details of the experiments on the synthetic dataset are added in Appendix G of the revised manuscript. We plan to release our synthetic data to the community after the paper review period.
>
> |Method|ACC|DP|EO|
> |----|----|----|----|
> |GRACE|60.41 $\pm$ 0.04|47.06 $\pm$ 4.20|47.01 $\pm$ 4.25|
> |FairDrop|59.05 $\pm$ 9.47|29.62 $\pm$ 12.88|29.66 $\pm$ 11.88|
> |NIFTY|56.38 $\pm$ 8.92|29.24 $\pm$ 12.16|29.16 $\pm$ 12.16|
> |FairAug|**60.42 $\pm$ 0.02**|43.40 $\pm$ 1.01|43.33 $\pm$1.00|
> |Graphair| 60.36 $\pm$ 0.02|**15.56 $\pm$ 9.01** |**15.57 $\pm$ 8.99**|
>
> >**Q2: From Figure 5, the proposed method seems to be very slow even on not-so-large datasets.**
>
> We agree that Graphair has a higher time complexity than other baselines during training. The reason is that the **baselines are non-learnable and use fixed strategies.** In contrast, **Graphair uses a learnable neural network** to automatically discover fairness-aware augmentations. Since training the automated augmentation model takes more time than using fixed augmentations, Graphair inevitably has a higher time complexity than the baselines. We will improve the efficiency of Graphair in future work.
>
> >**Q3: I suggest the authors use vector graphics for all figures in the paper.**
>
> We have updated all figures in the revised manuscript. We re-generate all figures as pdf files, and they should be vector graphics now.
>
> > Reference
>
> [1] Dai, E., & Wang, S. (2021, March). Say no to the discrimination: Learning fair graph neural networks with limited sensitive attribute information. In Proceedings of the 14th ACM International Conference on Web Search and Data Mining (pp. 680-688).
>
> [2] Dong, Y., Liu, N., Jalaian, B., & Li, J. (2022, April). Edits: Modeling and mitigating data bias for graph neural networks. In Proceedings of the ACM Web Conference 2022 (pp. 1259-1269).
>
> [3] Jiang, Z., Han, X., Fan, C., Liu, Z., Zou, N., Mostafavi, A., & Hu, X. (2022). Fmp: Toward fair graph message passing against topology bias. arXiv preprint arXiv:2202.04187.
>
> [4] Kose, O. D., & Shen, Y. (2022). Fair node representation learning via adaptive data augmentation. arXiv preprint arXiv:2201.08549.

---

> > ### Comment · Reviewer_fjjw · 2022-12-05
> > **thanks**
> >
> > I appreciate the authors' detailed response. I will keep my recommendation of acceptance.

---

### Decision · Program_Chairs · 2023-01-20

**Decision:**

Accept: notable-top-25%

**Justification For Why Not Higher Score:**

I think of it as the borderline between the spotlight and the poster. However, the reviewers' scores were better than I thought, so I suggested a spotlight. The method is simple and there are no surprising results, so I think it's not enough as an oral.

**Justification For Why Not Lower Score:**

see above

**Metareview: Summary, Strengths And Weaknesses:**

This paper proposes a data augmentation technique for fair graph representation learning.
All four reviewers agreed that the paper is very well written, easy to read, the problem is important, the idea is novel, and the experimental results are convincing and consistent. In the discussion, there was no disagreement in accepting the paper.
Some unclear parts or concerns such as large scale experiments were all resolved through rebuttal. I hope the authors check it again and make up for the lacking parts so that all of these concerns can be completely eliminated in the final version.

**Note From Pc:**

if the above contains the word "oral" or "spotlight" please see: "oral" presentation means -> notable-top-5% and "spotlight" means -> notable-top-25%. As stated in our emails, we are disassociating presentation type from AC recommendations